# Diabetic Polyneuropathy: New Strategies to Target Sensory Neurons in Dorsal Root Ganglia

**DOI:** 10.3390/ijms24065977

**Published:** 2023-03-22

**Authors:** Akiko Miyashita, Masaki Kobayashi, Takanori Yokota, Douglas W. Zochodne

**Affiliations:** 1Department of Neurology, Neurological Science, Tokyo Medical and Dental University, Tokyo 113-8519, Japan; 2Center for Brain Integration Research, Tokyo Medical and Dental University, Tokyo 113-8519, Japan; 3Department of Neurology, Nissan Tamagawa Hospital, Tokyo 158-0095, Japan; 4Division of Neurology and Department of Medicine, Faculty of Medicine and Dentistry, The Neuroscience and Mental Health Institute and The Alberta Diabetes Institute, University of Alberta, Edmonton, AB T6G 2G3, Canada

**Keywords:** diabetic polyneuropathy, dorsal root ganglia, noncoding RNAs, DNA/RNA heteroduplex oligonucleotide

## Abstract

Diabetic polyneuropathy (DPN) is the most common type of diabetic neuropathy, rendering a slowly progressive, symmetrical, and length-dependent dying-back axonopathy with preferential sensory involvement. Although the pathogenesis of DPN is complex, this review emphasizes the concept that hyperglycemia and metabolic stressors directly target sensory neurons in the dorsal root ganglia (DRG), leading to distal axonal degeneration. In this context, we discuss the role for DRG-targeting gene delivery, specifically oligonucleotide therapeutics for DPN. Molecules including insulin, GLP-1, PTEN, HSP27, RAGE, CWC22, and DUSP1 that impact neurotrophic signal transduction (for example, phosphatidylinositol-3 kinase/phosphorylated protein kinase B [PI3/pAkt] signaling) and other cellular networks may promote regeneration. Regenerative strategies may be essential in maintaining axon integrity during ongoing degeneration in diabetes mellitus (DM). We discuss specific new findings that relate to sensory neuron function in DM associated with abnormal dynamics of nuclear bodies such as Cajal bodies and nuclear speckles in which mRNA transcription and post-transcriptional processing occur. Manipulating noncoding RNAs such as microRNA and long-noncoding RNA (specifically *MALAT1*) that regulate gene expression through post-transcriptional modification are interesting avenues to consider in supporting neurons during DM. Finally, we present therapeutic possibilities around the use of a novel DNA/RNA heteroduplex oligonucleotide that provides more efficient gene knockdown in DRG than the single-stranded antisense oligonucleotide.

## 1. Introduction

Diabetes mellitus (DM) is a common chronic disease, and its complications are a major cause of morbidity and mortality worldwide [1]. Peripheral neuropathy is one of the major complications of diabetes mellitus, along with retinopathy and nephropathy. Diabetic polyneuropathy (DPN) is the most common form of a group of diabetes-related neuropathies that also include focal neuropathies and autonomic neuropathy. DPN is associated with sensory alterations in the distal parts of the extremities, with loss of sensation, impaired balance, and pain, all of which impair the quality of life of patients [2,3,4]. Optimal glycemic control is essential for the prevention of diabetic complications. Specific agents such as pregabalin and duloxetine have been developed to control neuropathic pain, and recent guidelines summarize position statements from the American Diabetes Association (ADA) and the American Academy of Neurology (AAN) [5,6]. Beyond these efforts, however, there has been limited progress in the development of molecular approaches that stop and reverse the progression of neuropathy. 

While DPN is likely a multifactorial disease, oxidative stress and its intersection with metabolic and vascular dysfunction induced by hyperglycemia have been considered to be a cardinal pathophysiological feature of diabetic complications [7]. A longstanding accepted target that impacts oxidative stress is aldose reductase (AR), which converts glucose to sorbitol as a rate-limiting enzyme in the polyol pathway. Despite extensive work, only Epalrestat is approved in Japan, but not in other jurisdictions, among several aldose reductase inhibitors (ARIs) which have been evaluated in clinical trials and may have efficacy in delaying the progression of DPN [8]. A Cochrane review has reviewed this history but is less supportive of their role [9]. Microcirculatory dysfunction associated with reduced vascularity and blood flow accompanied by a deficiency in angiogenic factors in peripheral nerves has also been regarded as a critical pathogenic factor for DPN [10]. However, whether microcirculatory changes are pathogenic or parallel developments and thereby less relevant to disease induction has been previously reviewed [11]. Schwann cell dysfunction is also considered an additional feature of peripheral nerve dysfunction in DPN, given the close support between Schwann cells and axons that is required [12,13]. A critical aspect of neuronal function in DM may relate to its impact on gene expression, given that the nucleus of the neuron executes the blueprint for the assembly of materials required for axon elongation and regeneration in response to cellular stress [14,15,16,17,18,19]. Elucidating how gene expression is specifically altered and influences neuron behavior of sensory neurons located in the dorsal root ganglia (DRG) under diabetic stress, will make possible to provide novel therapeutic targets to protect against or reverse DPN. 

In this review, we describe unique aspects of the pathogenesis of DPN that consider how gene expression in DRG neurons is involved in the development of DPN and how targeting DRG for the treatment of DPN may be relevant. Next, we focus on the intracellular signal transduction pathways in sensory neurons and introduce potential molecular interventions for DPN treatment that may influence the transcription of pathway signal molecules (Table 1). Specifically, we discuss the impact of noncoding RNAs that have recently attracted attention as having a role in the post-transcriptional modification of cellular protein expression. Integrating the knowledge of these molecular roles, we discuss the possibility that DM could be associated with abnormal dynamics of nuclear bodies in which mRNA transcription and post-transcriptional processing occur, ultimately promoting the degeneration of the distal axons of sensory neurons. Finally, we present future prospects for oligonucleotide therapeutics such as siRNA and ASO to control gene expression as effective knockdown approaches in DPN. As an initial proof of principle, we describe how a novel DNA/RNA heteroduplex oligonucleotide (HDO) developed in the Yokota laboratory, demonstrated functional impacts on the development of DPN, providing insights into the regulatory complex that may be pathogenic.

## 2. DRG Sensory Neurons in the Pathogenesis of Diabetic Polyneuropathy

Polyneuropathy exhibits a “stocking and glove” pattern of sensory, later motor, and often autonomic disturbances in the patient [2]. Patients with DPN manifest earlier sensory symptoms than motor symptoms, beginning in the distal extremities and progressing in the more proximal extremities. This course of progression suggests the development of a neurodegenerative process in which longer axons are preferentially damaged and undergo “dying-back.” This may be considered a form of neuronal cell body injury under toxic or metabolic alteration, leading to retrograde axon degeneration characterized by initial retraction of nerve terminals and perhaps, although not studied in humans, late loss of parent perikarya or cell bodies [51,52,53,54]. Evidence is accumulating that hyperglycemia and other insults related to diabetes can directly target the cell bodies of sensory neurons in DRG. Anatomically, sensory neurons are pseudounipolar neurons that have one axon divided into two branches and cell bodies surrounded by satellite glial cells in DRG. DRG have higher blood flow than nerve trunks and do not have a robust protective neurovascular barrier resulting in vulnerability to toxic circulating substances [55,56,57,58]. In chronic experimental DPN, axons are lost before neuronal cell bodies are lost, and pathological findings of atrophic axons due to declines in structural proteins such as neurofilament and α-tubulin are consistent with a down-regulation of their mRNA expression [14]. The process of degeneration in the sensory neurons may also be accompanied by declines in the expression of mRNAs for α Calcitonin gene-related peptide (CGRP), βCGRP, p75 neurotrophin receptor (NTR), the neurotrophic tyrosine kinase receptor (TrkA) and TrkC, but without elevation in mRNAs that normally rise after nerve injury such as Vasoactive intestinal peptide (VIP), galanin, Cholecystokinin (CCK), Pituitary adenylate cyclase-activating peptide (PACAP), growth-associated protein 43 (GAP43) and tα1-tubulin, which implies a specific degenerative process different from a nerve injury phenotype [15]. While DRG sensory neurons in diabetic mice showed down-regulation of mRNAs associated with growth proteins such as GAP43, they also demonstrated up-regulated mRNA expression of Phosphatase and tensin homolog deleted on chromosome 10 (PTEN) that inhibits phosphatidylinositol-3 kinase (PI3K)/phosphorylated protein kinase B (pAkt) growth pathway [16], and the neuroprotective factor heat shock protein 27 (HSP27) [30] (Figure 1). 

Given the uncertainty of whether perikaryal neuron damage precedes that of the axons or Schwann cells (SCs) distally in the nerve trunk, mechanisms of axonal degeneration, including Wallerian-like degeneration, are important to consider. Wallerian degeneration is the classic example of axonal degeneration, strictly characterized by self-digestion of the axons and myelin of an isolated distal nerve segment after nerve transection [52]. Slow Wallerian degeneration (Wld^S^) mutant mice express a chimeric nuclear protein, including enzymes involved in the biosynthesis of nicotinamide adenine dinucleotide, and these proteins protect injured axons from degeneration and delay the process of Wallerian degeneration for several weeks [59,60]. Wld^S^ mutant mice with superimposed experimental DM showed attenuation of nerve conduction velocity slowing and mechanical hypoalgesia [61], although the mechanism by which Wld^S^ protects against diabetic nerve damage is not yet clear. Beyond these considerations, pathological analysis of nerves in patients with DPN demonstrates not only axonal degeneration but also demyelination, indicating the involvement of SCs in the pathophysiology of DPN [12,62]. Hyperglycemia may induce perturbations in SC metabolism, ultimately leading to SC death, but diabetic mice do not show substantial demyelination beyond myelin thinning in long-term DM. AR is localized mainly in SCs [63], and several studies with rats found that SC damage is sensitive to ARIs, whereas pathological analysis of nerves in diabetic rats identified preventive impacts [64,65]. However, ARIs in patients with DPN have had limited benefits, and the significance and consequences of enhanced flux in the AR/polyol pathway in SCs continue to be debated.

## 3. Molecules That Impact Signal Transduction Pathways: Insulin, GLP-1, PTEN, HSP27, RAGE, CWC22, and DUSP1

Activation of neurotrophic effects in insulin/PI3K/pAkt signaling may be an important strategy to enhance the regenerative capacity of diabetic neurons in response to ongoing degeneration. Beyond the well-known actions of insulin in glucose homeostasis, the peptide has growth factor properties and is possessed of structural similarities with the Nerve Growth Factor (NGF), the “classical” neurotrophic factor involved in the growth, maintenance, proliferation, and survival of neurons [20]. Insulin receptors (IRs) are expressed in the peripheral nervous system, including sensory neurons, motor neurons, and Schwann cells [21,66]. Activation of insulin receptors, a receptor tyrosine kinase (RTK), induces conformational change and autophosphorylation and then recruits and phosphorylates receptor substrates such as IRS and Shc proteins, leading to activation of two downstream pathways. IRS activates the PI3K/pAkt pathway, likely its most important target, but Shc also activates the Ras/MAPK pathway [67,68]. The PI3K/pAkt pathway is highly conserved, and its activation is triggered by ligand binding to a range of receptors such as RTKs, Toll-like receptors (TLRs), and G protein-coupled receptors (GPCR). Akt is a significant effector of PI3K signaling in which recruited PDK1 phosphorylates Akt. Full activation of Akt mediates numerous cellular processes, including cell survival, growth, and proliferation. Insulin deficiency or resistance with downstream attenuation of PI3K/pAkt signaling in sensory neurons may have a role in the development of DPN. Intrathecal injections of low-dose insulin improved diabetic neuropathic changes in a long-term experimental type 1 diabetic rat model without an impact on blood glucose levels [22,23,24]. In type 2 diabetes, neurons might exhibit “insulin resistance” as reduced transduction downstream of the insulin receptor signaling. PI3K/pAkt activation in the DRG and sciatic nerve of type 2 DM mice was counteracted in response to an intrathecal injection of insulin associated with decreased DRG insulin receptor expression and up-regulation of JNK activity, a mediator of insulin resistance in other tissues [25]. High-dose insulin or repeated low-dose insulin resulted in down-regulated mRNA of the insulin receptor β-subunit, up-regulated GSK-3β which phosphorylates IRS at serine 332, impaired insulin signaling [69], and down-regulated pAkt. Subsequent challenges of insulin to support growth in these previously treated neurons were counteracted [70]. Overall, mechanisms of neuronal insulin resistance in type 2 diabetes may relate, at least in part, to PI3K/pAkt signaling deficiency. 

The neurotrophic effect of glucagon-like peptide-1(GLP-1) may be mediated by PI3K/pAkt. GLP-1 is an incretin hormone that is released from the L cells of the small intestine in a glucose-dependent manner and promotes insulin secretion through G-protein-coupled receptors expressed on islet β-cells in the pancreas. The GLP-1 receptors (GLP-1R) are expressed throughout the brain, and its signaling has been recognized to exert neuroprotective and neurotrophic effects in various experimental models of neurodegenerative diseases, including Alzheimer's disease, Parkinson’s disease, ALS, and ischemic brain injury [71]. The expression of GLP-1R was also confirmed in DRG sensory neurons of healthy and DM mice. A GLP-1 agonist enhanced neurite outgrowth in dissociated sensory neurons and improved neuropathic features of both type 1 and 2 DM models [26,27,28]. 

Molecules, including those classified as tumor suppressors, may brake intrinsic growth signals, including the PI3/pAkt pathway in neurons [72,73]. Phosphatase and tensin homolog deleted on chromosome 10 (PTEN) works as a negative regulator of the PI3K/pAkt signaling pathway [74,75,76,77]. PTEN knockdown enhances the outgrowth of neurons in both the central and peripheral nervous systems [29,78,79,80,81,82]. For example, its inhibition or knockdown using siRNA enhanced neurite outgrowth in dissociated sensory neurons and accelerated the regrowth of axons after sciatic nerve transection in rats [29]. In type 1 DM mice, the expression of PTEN is up-regulated in DRG sensory neurons, and its knockdown by nonviral delivery of siRNA enhanced nerve regeneration, including reinnervation of epidermal axons [16]. Ongoing work suggests PTEN knockdown can directly improve DPN.

Heat Shock Protein 27(HPS 27), belonging to the small molecular weight HSP family, is activated in response to cellular stresses such as heat shock and has been characterized in a protective role from disease states. HSP27 is constitutively expressed at a low level in the DRG neurons and dramatically up-regulated in sensory neurons [31,32]. It’s up-regulation likely contributes to neuron survival and protection, possibly through activation of the PI3/pAkt pathway by Akt phosphorylation of HSP27 [63,64,65,66]. HSP27 overexpression protected against loss of thermal sensation, mechanical allodynia, epidermal axon loss, and sensory conduction slowing in diabetic mice [15,30]. 

Chronic hyperglycemia generates advanced glycation endproducts (AGEs) as a result of the nonenzymatic glycation and oxidation of proteins and lipids. AGEs accumulate in systemic tissues and may contribute to diabetic complications by binding receptors for AGEs (RAGE) [83,84]. RAGE is expressed in a variety of cells, including neuronal and glial cells, and its activation enhances NADH-oxidase activity, resulting in the activation of NF-κB. The downstream impact of RAGE activation within sensory neurons is controversial, with data indicating involvement in the support and growth of neurons. RAGE activation promotes the outgrowth of dissociated sensory neurons through the NF-κB, JAK-STAT, and ERK pathways, and blockade of RAGE inhibits nerve regeneration after injury in mice [35,36]. However, RAGE null mice showed incomplete neuroprotection in experimental type 1 DM mice, and insulin administration provided only limited improvement of neuropathic indices of DPN when superimposed on RAGE depletion [37]. Further investigation may clarify whether activation of the AGE-RAGE signaling pathway contributes to DPN or has dual roles, given the findings to date addressing this pathway. 

As described earlier in this review, changes in intracellular signaling in DM may be associated with alterations in global gene expression within DRG sensory neurons (Figure 1). Shifts in transcript expression may be secondary to altered signaling or underlie the development of abnormal intracellular pathways. In a study of chronic type 1 experimental DM mice, mRNA expression profiles were analyzed by microarray that identified 91 up-regulated and 170 down-regulated mRNAs for a difference of at least 1.5-fold change among 28,869 mRNAs in DRGs [17]. Several up-regulated mRNAs, including *Cwc22* and *Dusp1*, may be promising therapeutic candidates. For instance, CWC22 is known to be an essential splicing factor that directly links the splicing machinery and the exon junction complex (EJC) by an escort of eIF4A3, a core EJC subunit, to the spliceosome. Its depletion results in global defects of pre-mRNA splicing and down-regulation of gene expression in vitro [38,39,40]. Inhibition of CWC22 in dissociated sensory neurons enhanced neurite outgrowth and improved features of neuropathy in chronic type 1 experimental DM [41]. In contrast, DUSP1 knockdown in adult sensory neurons in vitro impairs their plasticity during growth in dissociated sensory neurons and accelerates axon degeneration after axotomy, suggesting a role of DUSP1 in neuroprotection [42].

## 4. Targeting Key Players in Post-Transcriptional Regulation: miRNA and lncRNA

The regulation of gene expression occurs not only at the transcriptional level but also at the post-transcriptional level. The stability and distribution of RNA transcripts are regulated through post-transcriptional processing steps, including mRNA splicing, nuclear processing, RNA export, assembly, association with GW bodies, also known as processing bodies (P bodies), and RNA decay/degradation. GW/P bodies are cytoplasmic foci composed of ribonucleoprotein (RNP) granules involved in RNA degradation mediated by microRNAs (miRNA) [85,86,87,88,89]. We noted that DRG sensory neurons had up-regulation in the number of GW/P bodies under diabetic stress conditions, which might reflect increased degradation of specific RNAs mediated by miRNA in GW/P bodies [17]. miRNAs are small non-coding RNAs of 18–23 nucleotides that bind to complementary sequences in mRNAs, resulting in repressed gene expression at the post-transcriptional level. Precursor miRNAs are exported from the nucleus and associated with RNA-induced silencing complex (RISC) located in GW/P bodies. We found that 19 differentially expressed miRNAs (12 downregulated and 7 upregulated) were identified by microarray in DRGs of chronic type 1 DM mice [17]. Let-7i is one of the downregulated miRNAs belonging to let-7 family members that are highly conserved across species and might regulate the cell cycle, proliferation, and apoptosis [90,91]. DRG sensory neurons express let-7i, and exogenous let-7i promoted neurite growth and branching in dissociated sensory neurons, indicating trophic actions. Its replenishment in diabetic mice improved indices of neuropathy. In addition, a prominently upregulated miRNA in this model was miR-341, and its knock-down improved sensory nerve conduction and thermal sensitivity in diabetic mice, although miR-341 is only expressed in rodents. A significant up-regulation of miR-341 in the DRG of rats with chronic constriction injury is also reported [43]. Epigenetic alterations of GW/P bodies and miRNAs in DRG should be considered in the pathogenesis of DPN.

Long noncoding RNAs (lncRNAs), with ≥200 nucleotides, can also impose an additional level of post-transcriptional regulation [92,93]. Metastasis-associated lung adenocarcinoma transcript 1 (MALAT1) is one of the better-studied lncRNAs as being identified as a prognostic marker for non-small cell cancers [94]. It is expressed in most tissues but prominent in the nervous system, where it promotes synapse formation and neurite outgrowth through the activation of the ERK/MAPK signaling pathway in neurons [95,96]. MALAT1 localizes in nuclear speckles, which are subnuclear membrane-less structures enriched in pre-mRNA splicing factors, and regulates alternative splicing through interaction with serine/arginine splicing factors (SRSFs) [97]. For example, MALAT1 mediates the alternative splicing of protein kinase C to promote neuronal survival after injury [98]. MALAT1 was upregulated in the retina after optic nerve injury, and its knockdown reduced retinal ganglion cell survival through CREB signaling leading to retinal neurodegeneration, which suggests that MALAT1 has a neuroprotective role [99]. MALAT1 is upregulated in different diabetic complications, including retinopathy and kidney disease [100]. The upregulation of MALAT1 in diabetic retinopathy decreased the transcription of antioxidant defense genes in retinal endothelial cells, and its knockdown protected the retina from oxidative stress in diabetic retinopathy [101]. We recently identified up-regulation of MALAT1 in DRGs of diabetic mice. However, unlike the retina, phenotypic changes in diabetic mice induced by MALAT1 knockdown include additional deterioration of sensory nerve function, such as elevated nociceptive thresholds and further delayed conduction of the sciatic nerve accompanied by accelerated degeneration of axon terminals. Knockdown was also associated with nuclear structural changes characterized by the loss of nuclear speckles in DRG sensory neurons [44]. These changes suggested that MALAT1 might provide intrinsic neuroprotection in the setting of DM [92]. Additionally, a related nucleotide, lncRNA H19 has also been classified as an oncogene and promotes bladder cancer metastasis by associating with enhancer of zeste homolog 2 (EZH2) and inhibiting E-cadherin expression [102]. In the brain of diabetic rats, H19 is highly upregulated in rats with DM, which induces neuronal apoptosis mediated by Wnt signaling [45]. Furthermore, H19 expression is positively correlated with the expression of MALAT1 in blood samples of DPN patients, and these molecules are expected to be useful biomarkers for DPN [46].

Other up-regulated lncRNAs in DM include BC168687, uc. 48+, and NONRATT021972, confirmed in DRG of diabetic rats as well as blood samples in humans. BC168687 knockdown by siRNA suppressed the expression of TRPV1 and P2X7 receptors in DRG of rats, cytokines such as TNF-α and IL-1β in the serum, and nitric oxide (NO), which is generated by an increase in enzymatic activity of NO synthetase on diabetic DRG [103]. Thermal and mechanical hyperalgesia in DM rats were also improved [47,48]. Both uc.48+ and NONRATT021972 knockdown has also been reported to suppress excitatory transmission mediated by the P2X3 receptor in the DRG, and their knockdown reduced the level of tumor necrosis factor-α (TNF-α) in the serum, leading to alleviation of hyperalgesia in DM rats [49,50,104]. BC168687, uc 48+, and NONRATT021972 knockdown all improved sensory functions of diabetic pain models, but it is not clear whether their knockdown also improves pathological or electrophysiological evidence of neuropathy in DM.

## 5. Nuclear Bodies and Sensory Neurodegeneration in Diabetes

mRNA transcription and processing are localized to membraneless structures termed “nuclear bodies” within the nucleus. Nuclear bodies, such as nucleoli, Cajal bodies (CBs), and nuclear speckles, could offer a sequestered space that facilitates more efficient changes in gene expression [105]. CBs, which are made up of survival motor neuron (SMN) protein (also known as Gemini of CBs), control transcriptional activity in collaboration with nucleoli through the supply of small nuclear ribonucleoproteins (snRNPs) for the spliceosome that is required in pre-mRNA splicing [106,107,108,109,110]. Nuclear speckles, colocalized with CWC22, accumulate snRNPs and other non-snRNP protein splicing factors in a punctate nuclear localization pattern and provide a place to execute splicing [111]. Nuclear speckles also contain lncRNA MALAT1, which serves as a scaffold in nuclear speckles for RNA-binding proteins (RBPs), such as splicing factors. Nuclear speckles regulate transcriptional events involved in RNA processing and alternative splicing through the interaction of SR proteins [97,112,113]. We have found that CBs were increased in number but lost their colocalization with SMN proteins and formed abnormal aggregation of snRNPs in diabetic sensory neurons (Figure 2). This suggests a loss of recruitment of SMN proteins to CBs leading to dysfunction of spliceosome for pre-mRNA splicing [41]. The deficiency of SMN within CBs is known to be a key pathological finding of motor neurons in spinal muscular atrophy [114]. Furthermore, we recently found that the percentage of sensory neurons with nuclear speckles was moderately but significantly reduced in DM mice, and MALAT1 silencing induced a greater reduction in diabetic sensory neurons that accompanied terminal nerve loss (Figure 2) [44]. When taken together, sensory neurodegeneration in diabetes might be linked to splicing abnormalities accompanied by dynamic changes of nuclear bodies, similar to motor neuron death caused by the collapse of spliceosomal snRNP biogenesis and loss of SMN proteins in spinal muscular atrophy and perhaps ALS [115].

## 6. Therapeutic Strategies: Future Prospects of DRG-Targeting Gene Delivery

Efficient gene delivery and knockdown in DRG sensory neurons may be powerful tools in the future treatment of DPN (Figure 3). Oligonucleotide therapeutics such as siRNA and ASOs have been recently developed as promising strategies for a variety of neuromuscular diseases such as Duchenne muscular dystrophy, spinal muscular atrophy, and transthyretin familial amyloid polyneuropathy [116]. RNA interference (RNAi) is a natural cellular process that silences gene expression through the degradation of mRNA. As a relative of miRNA, siRNA is a short dsRNA that is produced by a ribonuclease enzyme III (RNase III) termed Dicer in the cytoplasm and elicits RNAi activity through cleavage of the target mRNA in association with Argonaute 2 (AGO2), a component of the RISC localized in GW/P bodies, processes that overall resemble miRNA-silencing [78,117]. ASOs are single-stranded nucleic acids synthesized to be complementary to target RNA sequences. ASOs are classified into two groups; RNase-dependent gapmer type and RNase-independent steric type. The gapmer type ASO is a phosphorothiate modified DNA gap sequence flanked by chemically modified oligonucleotides with high affinity and stability as a wing sequence such as 2′-*O*-methoxyethyl (MOE) and locked nucleic acid (LNA). The gapmer type ASO cleaves target RNA or natural antisense RNA by RNase H mainly in the nucleus to modulate gene expression. The steric type ASO hybridized the target pre-mRNA/mRNA to modulate splicing or inhibit mRNA translation. Although delivery of oligonucleotide therapeutics into the central nervous system is associated with some challenges of toxicity and delivery, small pharmacologic agents, including ASOs and siRNA, may have easier access to DRG neurons because the DRG lacks a blood-nerve barrier. Recently our laboratory has developed a new oligonucleotide therapeutic, double-stranded DNA/RNA heteroduplex oligonucleotide (HDO), which differs in structure from siRNAs of double-stranded RNA or ASOs of single-stranded DNA [118]. HDO is composed of DNA/locked nucleic acid (LNA) gapmer type ASO and its complementary RNA strand. Once an HDO enters the cell, the RNA strand of the duplex is selectively recognized and cleaved by RNase H, resulting in the release of the complementary strand from the parent ASO, efficient transport of the ASO into the nucleus to suppress the targeted mRNA (Figure 3). Toc-HDO, conjugated by cholesterol or alpha-tocopherol ligand to the complementary RNA sequence at the 5′ end, can much more effectively inhibit the target gene expression than a single-strand ASO in the central nervous system [119,120]. In addition, systemic administration of Toc-HDO can induce a highly efficient knockdown of DRG neuron mRNA compared to the parent single-stranded ASO with fewer treatment-related adverse effects, providing a new modality for DRG targeting strategy [121]. More than 90% silencing of MALAT1 expression in DRG of healthy mice did not impact pain behavior assessed by the acetone test, the Von Frey test, and the hot plate test [121]. Furthermore, the Toc-HDO targeting MALAT1, in the absence of DM alterations, had little direct toxicity to sensory neurons, but its inhibition in DPN promoted the further progression of neuropathic deficits, suggesting that the upregulation of MALAT1 in DPN may have a compensatory and protective role against diabetic stresses in DRG sensory neurons [44]. These specific findings indicated not only the efficacy of this new therapeutic approach in targeting but identified an unexpected and interesting role for a lncRNA not previously linked to DPN pathogenesis. Such a DRG-penetrating approach may be a very useful tool to both manipulate new therapeutic targets but also to elucidate critical features of DPN pathogenesis [122].

## 7. Conclusions

Targeting DRG sensory neurons in DPN may open new therapeutic avenues. A variety of mRNAs upregulated for neuroprotection or promotion of regenerative capacity associated with specific signal transduction pathways and post-transcriptional modifications in DM have been recently detected. Their knockdown produces phenotypic change, providing knowledge concerning pathophysiology and the possibility of candidates for DPN treatment. HDO is an effective oligonucleotide therapeutic tool to knock down the gene expression in DRG sensory neurons, which may be applicable for future explorations of DPN treatment. 

## Figures and Tables

**Figure 1 ijms-24-05977-f001:**
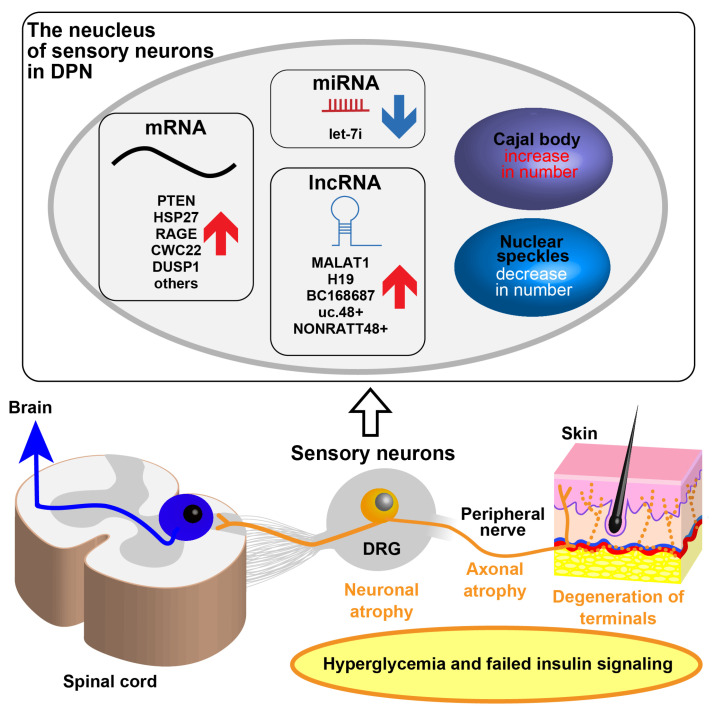
Schematic and simplified representation of pathogenetic mechanisms and alterations of gene expression within DRG during DPN. DPN is associated with DRG sensory neuronal atrophy and a reduced intraepidermal nerve fiber density (IENFD) due to a dying-back degeneration of distal axons. Gene expression changes in sensory neurons in DPN are accompanied by structural changes in nuclear bodies, which are essential for transcriptional activity.

**Figure 2 ijms-24-05977-f002:**
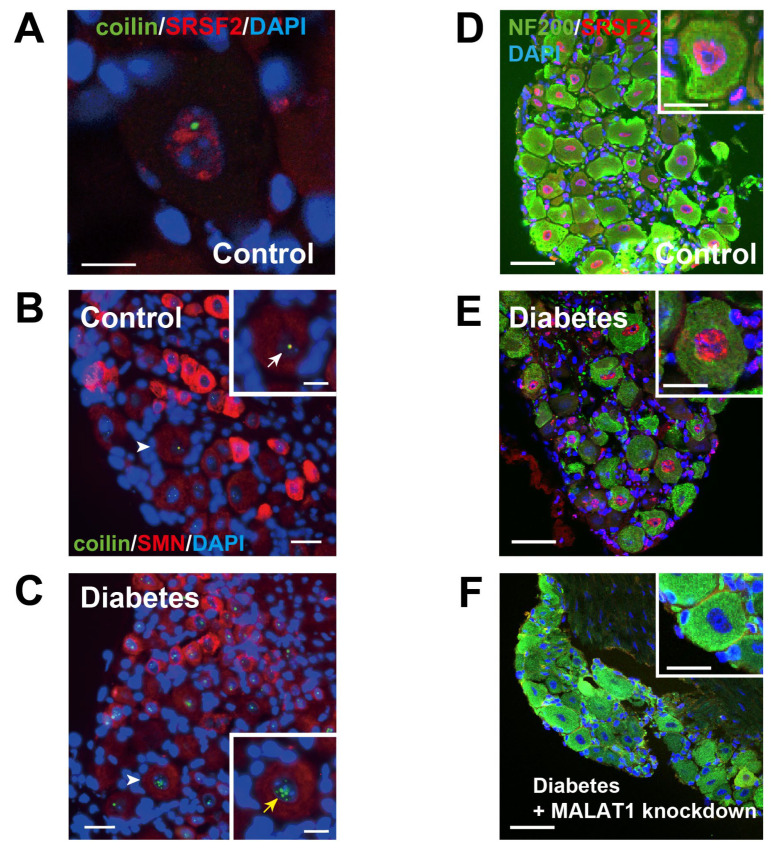
The structural alternations of nuclear bodies in DPN. In non-diabetic DRG sensory neurons, a single CB (coilin) was in contact with a nucleolus (DAPI), and nuclear speckles (SRSF2) were located in the interchromatin regions of the nucleoplasm in controls (**A**). Scale bar: 10 μm. In non-diabetic DRG sensory neurons, SMN protein was distributed throughout the cytoplasm and as nuclear foci (**B**); Scale bars: 20 μm, 10 μm in insets. SMN nuclear foci that collaborate on the assembly of snRNPs were localized within CBs (coilin) in controls (middle row, white arrows) but numerous CBs lost co-localization with SMN nuclear foci (bottom row, yellow arrow) in diabetic nuclei (**C**). Scale bars: 20 μm, 10 μm in insets. Arrowheads indicate sensory neurons magnified in the insets. SRSF2 is expressed in the nuclear speckles, where MALAT1 is localized, in the DRG sensory neurons in non-diabetic control mice. Anti-neurofilament 200 (NF200) is a marker of large and small myelinated neurons (**D**). DRG neurons with SRSF2-positive nuclear speckles are moderately reduced in the diabetic mice (**E**); They are further decreased in diabetic mice with MALAT1 silencing (**F**). Scale bars = 50 μm, and 20 μm, in insets. (**A**–**C**) were adapted with permission from Ref. [41]. 2017 Zochodne, D.W. and (**C**–**F**) from Ref. [44]. 2022 Yokota, T.

**Figure 3 ijms-24-05977-f003:**
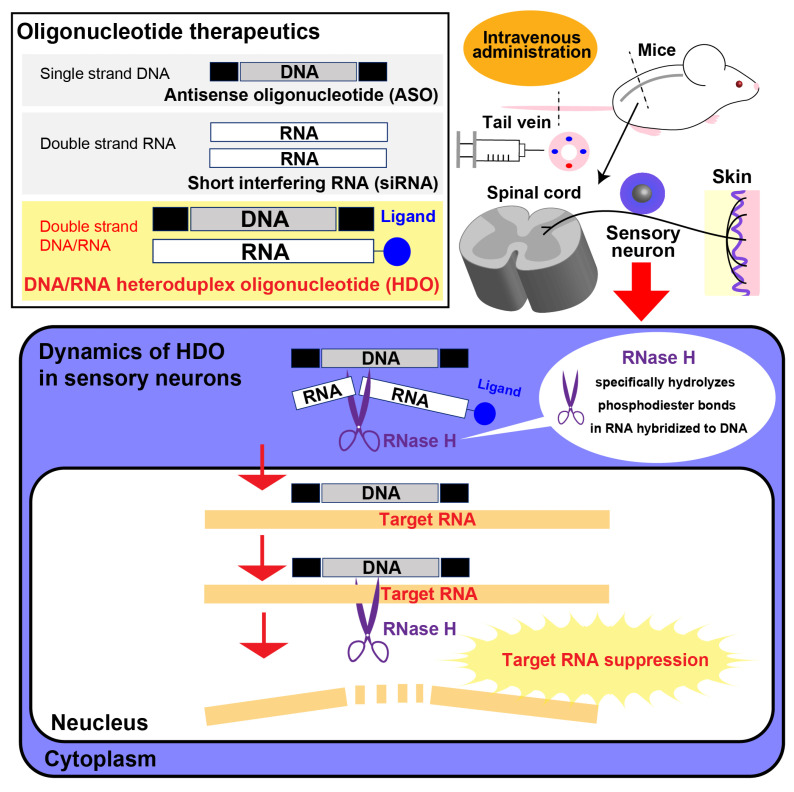
Schematic representation of oligonucleotide therapeutics and their mechanisms. HDO suppresses target RNAs. It is a double-stranded artificial functional nucleic acid consisting of a DNA strand as the main strand and an RNA complementary to the main strand, which is called a “gapmer” nucleic acid (LNA). This part is recognized by RNase H, an enzyme that degrades RNA in the cell, and the complementary strand RNA is cleaved. The resulting single main strand binds to the target RNA, and RNase H again cleaves the target RNA to exert its gene suppression effect.

**Table 1 ijms-24-05977-t001:** A selective list of therapeutic molecular candidates in DRG sensory neurons for DPN in this review.

Therapeutic Candidates		Regulation in Diabetes	Neuronal Function Related to Pathophysiology of DPN	Reference
PI3/pAkt pathway	*Insulin*	Deficiency/resistance	Insulin receptors are expressed in DRG neurons. Insulin deficiency or resistance may promote DPN via attenuation of PI3/pAkt signaling in DRG of DM mice. Insulin administration improved neuropathic deficits in DM mice	[20,21,22,23,24,25]
	*GLP-1*	Deficiency	GLP-1R was expressed in DRG neurons. GLP-1 agonist enhanced neurite outgrowth in dissociated sensory neurons and improved neuropathic deficits in DM mice.	[26,27,28]
	*PTEN*	Up-regulated	PTEN is a negative regulator of PI3/pAkt pathway. PTEN knockdown enhanced neurite outgrowth in dissociated sensory neurons and nerve regeneration in injured axon in DM mice.	[16,29]
	*HSP27*	Up-regulated	HSP27 is upregulated in neurons and promotes axonal growth after nerve injury. HSP27 overexpression protected against sensory dysfunction and nerve terminal loss in DM mice, possibly via Akt phosphorylation by HSP27.	[30,31,32,33,34]
*RAGE*		Up-regulated	RAGE activation promoted the outgrowth of dissociated sensory neurons in mice, and RAGE blockade inhibits nerve regeneration after injury, but showed incomplete neuroprotection in DM mice.	[35,36,37]
*CWC22*		Up-regulated	CWC22 is an essential splicing factor. Its knockdown enhanced neurite outgrowth in dissociated sensory neurons and improved sensory nerve function in DM mice.	[17,38,39,40,41]
*DUSP1*		Up-regulated	DUSP1 knockdown impaired outgrowth in dissociated sensory neurons and enhanced axon degeneration after nerve cut, but its neuroprotection in DM mice is still unknown.	[17,42]
miRNA	*let-7i*	Down-regulated	Exogeneous let 7i promoted neurite outgrowth in dissociated sensory neurons and its replenishment in DM mice improved neuropathic deficits.	[17]
	*miR-341*	Up-regulated	miR-341 was upregulated in DRG of rats with constriction injury. Its knockdown improved sensory nerve functions in DM mice. miR-341 is only expressed in rodents.	[17,43]
lncRNA	*MALAT1*	Up-regulated	MALAT1 knockdown induced a further exacerbation of neuropathic deficits along with loss of nuclear speckles in diabetic mice.	[44]
	*H19*	Up-regulated	H19 overexpression induced neuronal apoptosis via Wnt signaling in DM rats. H19 expression was positively correlated with MALAT1 expression in blood samples of DPN patients.	[45,46]
	*BC168687*	Up-regulated	BC168687 knockdown alleviated hyperalgesia via inhibition of TRPV1 receptor, P2X_7_, cytokines and nitric oxide in DRG of DM rats.	[47,48]
	*uc.48+*	Up-regulated	Uc.48+/NONRATT021972 knockdown alleviates hyperalgesia associated with inhibition of the excitatory transmission via P2X_3_ receptor and reduced the level of TNF-α in DM rats.	[49,50]
	*NONRATT021972*	Up-regulated		

## Data Availability

Not applicable.

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
