# Peer review of "Diabetic Polyneuropathy: New Strategies to Target Sensory Neurons in Dorsal Root Ganglia"

_ijms, 2023, doi:10.3390/ijms24065977_

Round 1

Reviewer 1 Report

  The topic of the article is relevant, interesting and original. The article systematizes knowledge on new strategies targeting sensory neurons in dorsal root ganglia in patients with diabetic polyneuropathy.

The work is well written.
The text is clear and easy to read.
The conclusions are consistent with the evidence and arguments presented. 

Author Response

Thank you for considering our previous submission of the manuscript. We have reviewed the comments of the referees and revised the paper accordingly. Below we indicate how we have responded to the comments provided:

Response to Reviewers and the Associate Editor

Blue indicates the parts that we changed according to Reviewer 2. Green indicates the parts that we changed according to Reviewer 3.

To Reviewer 1

Thank you very much for your advice .

Comment: The topic of the article is relevant, interesting and original. The article systematizes knowledge on new strategies targeting sensory neurons in dorsal root ganglia in patients with diabetic polyneuropathy. The work is well written.

The text is clear and easy to read. The conclusions are consistent with the evidence and arguments presented.

Response: Thank you for these comments.

Reviewer 2 Report

To the authors, 

In my opinion, the review contains interesting information ,is well written , well documented and properly discussed; moreover included tables and schematic representation which facilite understanding of the text.

On the basis of the above comments i am please to recommend this review for publication in IJMS. Only a minor aclaration it must be  do it respect to legend of Figure2.

In the figure 2d, antibodies used are SRF2 and NF200, but the last one is not mencioned in the legend. I supposed NF200 is neurofilament 200 and it is used for detecting neuronal bodies. I think it must be added to the legend.

Moreover, in figure2b, i do not understand why the author explain that exists colocalization  between SMN and coilin. 

Author Response

Thank you for considering our previous submission of the manuscript. We have reviewed the comments of the referees and revised the paper accordingly. Below we indicate how we have responded to the comments provided:

Response to Reviewers and the Associate Editor

Blue indicates the parts that we changed according to Reviewer 2. Green indicates the parts that we changed according to Reviewer 3.

To Reviewer 2

Thank you very much for your advice. We agree with the four issues raised and we have  incorporated them into R1 version blue color).

Comment: In my opinion, the review contains interesting information ,is well written , well documented and properly discussed; moreover included tables and schematic representation which facilite understanding of the text.

On the basis of the above comments i am please to recommend this review for publication in IJMS. Only a minor aclaration it must be  do it respect to legend of Figure2.

In the figure 2d, antibodies used are SRF2 and NF200, but the last one is not mencioned in the legend. I supposed NF200 is neurofilament 200 and it is used for detecting neuronal bodies. I think it must be added to the legend.

Moreover, in figure2b, i do not understand why the author explain that exists colocalization  between SMN and coilin.

Response: Thank you for this comment. We added the description in the legend of Fig. 2D (page 9, Lines 339-340 of the R2 version, in blue color) as follows: “Anti-neurofilament 200 (NF200) is a marker of large and small myelinated neurons.”

We also added the description of  the description in the legend of Fig. 2b (page 9, Lines 335 of the R2 version, in blue color) as follows: “that collaborate on the assembly of snRNPs.” 

Reviewer 3 Report

Journal: International Journal of Molecular Science

Title: Diabetic Polyneuropathy: New strategies to target sensory neurons in dorsal root ganglia

The authors review the possible causes for diabetic polyneuropathy with a focus on dorsal root ganglia and discuss possible pathways to intervene and to alleviate it. They concentrate mainly on the role of gene delivery to dorsal root ganglia.

The data are interesting and highlight some new possible ways for future treating of diabetic neuropathy.

There are only a very few points for corrections.

·        Line 44: citation of refs "[5][6]", but in line 62 "[12,13]", same for line 203 and 356. Please write it coherently.

·        Line 54: "efficacy in y delaying" ?

·        Line 104: "[24][25][26][27]In chronic", but in line 64-65 "[14-19]", same for line 313. Please write it coherently.

·        Line 130: "express a chimeric nuclear proteins including" either singular or plural.

·        Line 337: "in controls (top row, white" It is not top row but middle row.

·        Line 349: "neurological diseases such as Duchenne muscular dystrophy" Duchenne muscular dystrophy is not a neurological disease. The missing dystrophin is a structural protein under the cell membrane. It causes dead of muscle cells. Only a small part of DMD patient have cognitive impairments because of mutations in the brain isoform of dystrophin.

·        Line 363: "or inhibit miRNA translation" really miRNA translation?

·        Line 406 – 408: "his section may be divided by subheadings. It should provide a concise and precise description of the experimental results, their interpretation, as well as the experimental conclusions that can be drawn." Please delete it.

·        Line 444: "¥Scott," correct name?

·        Line 560:"RONG, L.L. Antagonism of RAGE", upper and lower case and Rong is not the only author, LL Rong, W Trojaborg, W Qu, K Kostov…

Author Response

Thank you for considering our previous submission of the manuscript. We have reviewed the comments of the referees and revised the paper accordingly. Below we indicate how we have responded to the comments provided:

Response to Reviewers and the Associate Editor

Blue indicates the parts that we changed according to Reviewer 2. Green indicates the parts that we changed according to Reviewer 3.

To Reviewer 3

Thank you for these comments . We have incorporated changes into the R3 version green color).

Comment: The authors review the possible causes for diabetic polyneuropathy with a focus on dorsal root ganglia and discuss possible pathways to intervene and to alleviate it. They concentrate mainly on the role of gene delivery to dorsal root ganglia.

The data are interesting and highlight some new possible ways for future treating of diabetic neuropathy.

There are only a very few points for corrections.

  • Line 44: citation of refs "[5][6]", but in line 62 "[12,13]", same for line 203 and 356. Please write it coherently.

  • Line 54: "efficacy in y delaying" ?

  • Line 104: "[24][25][26][27]In chronic", but in line 64-65 "[14-19]", same for line 313. Please write it coherently.

  • Line 130: "express a chimeric nuclear proteins including" either singular or plural.

  • Line 337: "in controls (top row, white" It is not top row but middle row.

  • Line 349: "neurological diseases such as Duchenne muscular dystrophy" Duchenne muscular dystrophy is not a neurological disease. The missing dystrophin is a structural protein under the cell membrane. It causes dead of muscle cells. Only a small part of DMD patient have cognitive impairments because of mutations in the brain isoform of dystrophin.

  • Line 363: "or inhibit miRNA translation" really miRNA translation?

  • Line 406 – 408: "his section may be divided by subheadings. It should provide a concise and precise description of the experimental results, their interpretation, as well as the experimental conclusions that can be drawn." Please delete it.

  • Line 444: "¥Scott," correct name?

  • Line 560:"RONG, L.L. Antagonism of RAGE", upper and lower case and Rong is not the only author, LL Rong, W Trojaborg, W Qu, K Kostov…

Response: Thank you for this comment. I've corrected them as follows:

  • "[5,6]" (page 1, Lines 44 of the R3 version, in green color)
  • "[29,30]" (page 5, Lines 131 of the R3 version, in green color)
  • "[78,117]" (page 10, Lines 355 of the R3 version, in green color)

  • "in delaying" (page 2, Lines 54 of the R3 version, in green color)

  • "[24-27]" (page 4, Lines 101 of the R3 version, in green color)
  • "[106-110]" (page 8, Lines 310 of the R3 version, in green color)
  • "[90,112,113]" (page 8, Lines 316 of the R3 version, in green color)

  • "a chimeric nuclear protein" (page 5, Lines 128 of the R3 version, in green color)

  • "middle row" (page 9, Lines 334 of the R3 version, in green color)

  • "neuromuscular disease" (page 9, Lines 348 of the R3 version, in green color)

  • "inhibit mRNA translation" (page 10, Lines 362 of the R3 version, in green color)

  • We deleted "his section may be divided by subheadings. It should provide a concise and precise description of the experimental results, their interpretation, as well as the experimental conclusions that can be drawn." (page 11, Lines 405 of the R3 version, in green color)

  • "Scott" (page 12, Lines 441 of the R3 version, in green color)

  • "Rong, L.L.; Trojaborg, W.; Qu, W.; Kostov, K.; Yan, S.D.; Gooch, C.; Szabolcs, M.; Hays, A.P.; Schmidt, A.M." (page 14, Lines 557 of the R3 version, in green color)

We have also taken the opportunity to make some minor improvements in the writing (highlighted in Red).
